# Therapeutic Applications of Solid Dispersions for Drugs and New Molecules: In Vitro and In Vivo Activities

**DOI:** 10.3390/pharmaceutics12100933

**Published:** 2020-09-30

**Authors:** Verônica da Silva Oliveira, Amanda Silva de Almeida, Ingrid da Silva Albuquerque, Fernanda Ílary Costa Duarte, Bárbara Cristina Silva Holanda Queiroz, Attilio Converti, Ádley Antonini Neves de Lima

**Affiliations:** 1Postgraduate Program in Pharmaceutical Sciences, Health Sciences Center, Federal University of Rio Grande do Norte, Natal-RN 59012-570, Brazil; veronicasoliver47@gmail.com (V.d.S.O.); fernandailary@gmail.com (F.Í.C.D.); barbaracqueiroz@gmail.com (B.C.S.H.Q.); 2Department of Pharmacy, Universidade Federal do Rio Grande do Norte, Natal-RN 59012-570, Brazil; amandasdealmeida@gmail.com (A.S.d.A.); ingridmariaalbuquerque@gmail.com (I.d.S.A.); 3Department of Civil, Chemical and Environmental Engineering, Pole of Chemical Engineering, Genoa University, I-16145 Genoa, Italy; converti@unige.it

**Keywords:** solid dispersions, drug delivery, biological applications

## Abstract

This review aims to provide an overview of studies that address the use, in therapeutic applications, of solid dispersions (SDs) with biological activities in vitro and/or in vivo mainly made up of polymeric matrices, as well as to evaluate the bioactive activity of their constituents. This bibliographic survey shows that the development of solid dispersions provides benefits in the physicochemical properties of bioactive compounds, which lead to an increase in their biological potential. However, despite the reports found on solid dispersions, there is still a need for biological assay-based studies, mainly in vivo, to assist in the investigation and to devise new applications. Therefore, studies based on such an approach are of great importance to enhance and extend the use of solid dispersions in the most diverse therapeutic applications.

## 1. Introduction

Many drug candidates have low aqueous solubility, which can make their oral absorption inadequate. According to the literature, approximately 40% of marketed drugs are poorly soluble in water, as are, according to the Biopharmaceutical Classification System, about 40–90% of new drug candidates [1,2,3,4].

The slow dissolution rate and low solubility of some drugs lead to unpredictable bioavailability, non-reproducible clinical response or treatment inefficiency due to low therapeutic plasma levels; therefore, their use must be optimized using formulation strategies capable of improving their administration [5].

Several technological strategies have been developed and employed to circumvent this situation such as complexation with cyclodextrins, particle size reduction by micronization, crystal development, nanotechnology [5], salt formation and solid dispersions (SDs) [6,7]. However, some of these strategies have shown disadvantages including the development of active forms in vivo, high execution cost and considerable levels of toxicity. Some of these limits can be overcome by employing solid dispersions, a viable, well-established and widely-used strategy to increase the dissolution rate and solubility of poorly water-soluble drugs [4,6,7].

SDs can be defined as molecular mixtures of drugs that are not soluble in hydrophilic carriers, which exhibit a drug release profile driven by the polymer properties [8]. The so-called first-generation crystalline SDs, which use crystalline carriers such as urea and sugars including sucrose, dextrose and galactose, have the disadvantage of high thermodynamic stability that prevents a quick release of drugs [9]. In second-generation SDs drugs are dispersed in usually polymeric amorphous carriers [8], while the carriers used in third-generation SDs are surfactant or a mixture of amorphous polymers and surfactants [10].

The use of hydrophilic polymeric matrices to develop SDs for the release of commercial drugs or new drugs candidates has been shown to be a promising alternative able to improve their pharmacokinetic properties [7].

This review deals with SDs, mainly those belonging to the second-generation class that use polymers as carriers. It describes the main in vitro and in vivo activities (Figure 1) of SDs having different compositions and various preparation methods, intending to innovate and enhance the release of poorly water-soluble drugs, as well as improving the biological activities of the loaded bioactive compounds.

## 2. Methods

This literature search was conducted through the ScienceDirect and PubMed specialized databases, using different combinations of the following keywords: “amorphous solid dispersions” or “in vitro solid dispersions” or “in vivo solid dispersions”. Studies performed in the last 10 years in vitro and in vivo were included, in which the biological action and use of solid dispersions to improve the biological performance of substances were investigated.

## 3. In Vitro Study of Solid Dispersions in Polymeric Matrices

This section deals with solid dispersions (SDs) prepared to improve the properties and release of poorly soluble drugs and drug candidates of natural or synthetic origin. The preparation and use of SDs have been reported in several in vitro studies, which have been numerically quantified and classified in Figure 2 based on the biological activities of their active compounds, while the main information on these studies is summarized in Table 1.

### 3.1. Anticancer Activities of Solid Dispersions

To improve the solubility and bioavailability of niclosamide, an anthelmintic with promising anticancer activity, a SD of its mixture with the octenylsuccinate hydroxypropyl phytoglycogen (OHPP) biopolymer was prepared by spray drying. In vitro tests by the 3-(4,5-dimethylthiazol-2-yl)-2,5-diphenyltetrazolium bromide (MTT) assay provided half inhibitory concentration (IC50 values) of SD against HeLa (cervical carcinoma), PC-3 (prostate cancer) and A549 (lung cancer) cells of 0.92, 0.68 and 1.18 μg/mL, while those of niclosamide alone were 1.82, 5.93 and 1.46 μg/mL, respectively. These results demonstrated that niclosamide-OHPP SD showed higher cytotoxicity than DMSO-assisted niclosamide solutions, mostly on PC-3 cell line, although OHPP, as the solubilizer, did not contribute to cytotoxicity. In addition, the analysis on OHPP alone is missed [11].

Although paclitaxel (PTX) is a natural compound with powerful antitumor activity against lung, breast and ovarian cancers, its low water solubility poses a challenge for the manufacture and administration of its formulations [12]. There are currently two commercial forms, Taxol^®^ and Abraxane^®^, which; however, have a high rate of hypersensitivity and side effects.

Considering these drawbacks, Xie and Yao [13] incorporated PTX with OHPP in the form of solid dispersion (PTX-OHPP SD) by spray drying, whose in vitro antitumor activity was tested by the MTT method against HeLa, PC-3 and A549 cells and compared to that of PTX alone. Cytotoxicity was dose dependent and OHPP alone was not cytotoxic to cells, with an inhibition rate lower than 15% at the highest concentration (93.75 μg/mL). The IC50 values of DMSO-assisted PTX solutions against the above tumor cells were 0.02, 0.08 and 0.03 μg/mL, respectively, while PTX-OHPP SD demonstrated the greatest efficacy against cancer cells, with values of 0.008, 0.017 and 0.002 μg/mL, respectively.

In another study conducted by Choi et al. [14], the goal was to develop PTX-containing SDs. Several polymers were tested to increase the drug solubility and, subsequently, to select the best polymer to prepare SDs by the solvent evaporation method. Solubility and dissolution tests allowed to select a SD composed of D-α-tocopheryl polyethylene glycol 1000 succinate (TPGS), polyvinylpyrrolidone/vinyl acetate (PVP/VA), Aerosil^®^200 and PTX (SD4) and another containing PTX, TPGS and Aerosil^®^200 (SD9), whose antitumor activities were assessed, at drug concentrations of 0.1, 1, 5, 10 and 20 µg/mL, against breast cancer cells (BT-474, MCF-7 and SK-BR-3) and non-tumor cells by the MTT assay. Whereas both SDs were nontoxic to normal cells, at a dose of 20 μg/mL the cytotoxicity of SD4 and SD9 was 3.0- and 2.8-fold higher than that of PTX alone against BT-474 cells, 4.0- and 4.1-fold against MCF-7 cells and 5.6- and 5.0-fold against SK-BR-3 cells. These results show greater cytotoxicity of SD4 and SD9 compared to PTX alone at the same concentration of 20 µg/mL.

In order to improve the solubility and absorption of chrysin, a potent inhibitor of breast cancer resistance protein, Lee et al. [15] have developed SD formulations with hydrophilic carriers, including polyoxyethylene (4) lauryl ether (Brij^®^L4), using the solvent evaporation method. To evaluate the cytotoxicity of chrysin alone and SD in the concentration range of 5–160 μM, the MTT assay against HT29 (human colorectal carcinoma) cells was used. SD displayed a 50% cytotoxic concentration (CC50) of 26.3 μM, while pure chrysin showed low activity even at the highest concentration tested.

Despite its wide variety of bioactive properties, Curcumin (CM), the main bioactive polyphenolic compound in the rhizomes of *Curcuma longa* L., has low solubility in water and; therefore, a limited bioavailability, which reduces its applicability. Based on these limitations, SDs of curcumin were prepared and tested for cytotoxic activity against tumor cells, namely MCF-7, NCIH460 (non-small cell lung carcinoma), HeLa and HepG2 (carcinoma hepatocellular) cells and non-tumor (PLP2) cells [16]. SDs showed cytotoxicity to all tumor cell lines tested, with concentrations causing 50% cell growth inhibition (GI50) of 207 ± 4, 208 ± 3, 259 ± 8 and 240 ± 7 μg/mL against NCIH460, HeLa, HepG2 and MCF-7, respectively, whereas no cytotoxicity was detected to non-tumor cells, with GI50 > 400 μg/mL. The improved CM activity was attributed to its association with copolymer surfactants such as Poloxamer 407 used to prepare SDs, which increased the dissolution rate and bioavailability of this hydrophobic drug.

In another study on CM, SD of zinc(II)-curcumin complex was developed using polyvinylpyrrolidone (PVP K30) [17]. Metal-curcumin complexes are known to exert anticancer activity, but they have low bioavailability and poor solubility in water, which hinder oral formulations. To evaluate the in vitro activity, the MTT assay was used in hepatocellular carcinoma cell lines (HepG2 and SK-HEP1), using different concentrations of SDs containing pure CM (CM-SD) or Zn(II)-curcumin complex (ZnCM-SD). Treatment with CM-SD significantly reduced the viability of HepG2 and SK-HEP1 cells within 48 h, with IC50 of 17.73 and 57.31 nM, respectively, while cells treated with ZnCM-SD exhibited after the same time IC50 of 38.52 and 10.41 nM, respectively, in a dose-dependent manner.

Xiong et al. [18] prepared telaprevir-containing SDs by the co-grinding technique using hydroxypropyl methylcellulose (HPMC), PVP K30 and polyethylene glycol (PEG) 6000 as polymers. SDs were characterized and their cytotoxicity was evaluated in vitro by the MTT method against HepG2 and normal liver cells (LO2) at drug concentrations between 0.1 and 30 μM. Although the viability of HepG2 cells gradually decreased as the drug concentration increased, it was almost the same in the presence of SDs and pure drug; furthermore, the efficacy of telaprevir was maintained even after the addition of polymers and no systems developed were toxic to normal liver cells.

Considering that uterine carcinoma still affects several women worldwide and is responsible for the highest rate of female malignancy [19], a study developed by Jiang et al. [20] sought to evaluate the anticancer activity of *Angelica gigas* Nakai, a medicinal herb containing coumarins, essential oils and polyacetylene that has several biological activities. However, its main components have a hydrophobic character that affects its application in formulations. To overcome this difficulty, these authors developed a SD of the above herb using the Soluplus^®^ polymer and subjected it to shear by hot-melt extrusion. The MTT method was used for the in vitro antiproliferative activity and the developed systems were tested at concentrations of 50, 100 and 200 µg/mL against HeLa cells and normal cells (HEK 293 lineage). All formulations reduced cell viability, but the hot-melt extruded powder led to the lowest HeLa cells viability after 24 h. When these cells were treated for 48 h with the SD, there were a considerable reduction in viability and a higher cytotoxicity compared to the others, with only 17.37% of viability being observed at the concentration of 200 µg/mL. On the other hand, none of the systems tested showed cytotoxicity in 24 or 48 h to HEK 293 cells.

The research developed by Guo et al. [21] aimed at obtaining a SD using a methacrylic acid polymer Eudragit S-100 in association with berberine hydrochloride (HB), which is an isoquinoline-derived alkaloid found widely in the root and stem bark of numerous medicinal plants. Previous in vitro studies did in fact show that berberine decreases the viability of breast cancer cells, lung and hepatocellular carcinoma [22,23]. However, the development of formulations using HB is hampered by the low bioavailability of this compound that causes irritation to the gastrointestinal tract after oral administration. In this context, a SD made up of HB and Eudragit S-100 was prepared by the solvent evaporation method and its antitumor activity evaluated by the MTT method against SW480, HCT116 and Caco-2 colon cancer cells, over time periods of 12, 24 and 48 h, at HB equivalent concentrations from 0 to 100 µM [21]. The IC50 values obtained after 24 h treatment with HB alone against the above cells were 88.64, 67.83 and 89.78 µM, respectively, while those of SD were 52.96, 42.81 and 37.68 µM, respectively. After 48 h treatment, both showed better results, with pure HB IC50 values of 43.10, 34.13 and 32.82 µM and SD IC50 values of 28.15, 22.06 and 15.39 µM against the same cells in the above order. These results showed that cell viability of these strains suffered a dose and time dependent reduction in the presence either of HB or SD.

Kumar et al. [24] developed binary and ternary SDs incorporating IIIM-290, a synthetic chromone alkaloid acting as a potent inhibitor of CDK2/A and CDK9/T1 kinases, whose poor water solubility causes it to be used in high doses. SDs were prepared by solvent casting and evaporation method using PVP K30, PEG-Polypropylene glycol(PPG)-PEG or xanthan gum as polymers. In in vitro cytotoxicity tests against Ehrlich ascites carcinoma cells, IIIM-290 alone and the best drug-containing SD prepared with PVP K30 showed IC50 values of 1.10 ± 0.05 and 1.35 ± 0.02 µg/mL, respectively.

The results summarized in this section demonstrated that the use of SDs not only allowed to increase the solubility and dissolution of the bioactive substances, but also to improve their in vitro antitumor activity, as evidenced mainly by a IC50 reduction; therefore, it can be deduced that this improvement derives from the increase in the solubility of active compounds when present in the dispersed systems. Furthermore, it is important to note that SDs were obtained for a large number of active compounds and that toxicity tests were performed on different cell types, which demonstrates their potential as systems for the release of lipophilic compounds to fight a variety of cancers.

### 3.2. Antiparasitic Activity of Solid Dispersions

#### 3.2.1. Antichagasic Activity of Solid Dispersions

Chagas disease is an infection that affects around eight million people around the world, mainly in Latin America, which is treated with nifurtimox and benznidazole (BNZ). Although the latter is preferable because of fewer adverse effects than the former, it has low solubility, which limits its effectiveness in treating the disease [25].

One of the first studies developed to obtain BNZ-containing SDs was conducted by Lima et al. [26] with the aim of improving the solubility of this drug. In this work, the use of PEG 6000 and PVP K30 as hydrophilic polymer carriers led to an increase in the solubility of BNZ in aqueous medium. The interaction of BNZ with polymers was evaluated by physicochemical characterization and in vitro dissolution tests. The dissolution profile of all SDs was significantly better than that of BNZ alone, thus demonstrating the solubilizing effect of PEG and PVP. Another work published by the same group [27] aimed to develop SDs with HPMC and β-cyclodextrin inclusion complexes to increase the drug solubility and dissolution.

In a study by Fonseca-Berzal et al. [28], BNZ-containing SDs were prepared by the freeze-drying technique using sodium deoxycholate (NaDC) as surfactant and low-substituted hydroxypropyl cellulose (HPC) as polymer, whose trypanocidal activity was tested in vitro against axenic cultures of *Trypanosoma cruzi*. The results showed greater activity of the 1:3 BNZ:NaDC solid dispersion, with IC50 values of 33.92 ± 6.41 µM for epimastigote and 0.40 ± 0.05 µM for amastigote, compared to pure BNZ (IC50 of 52.95 ± 2.22 and 1.8 ± 0.66 µM, respectively).

To potentiate the effect of the same drug, Simonazzi et al. [25] prepared a BNZ-containing SD using Poloxamer 407 (P407) as polymer, whose in vitro activity was tested by the MTT method against *T. cruzi* (Tulahuén lineage) in the epimastigote phase at a concentration of 4 × 107 cells/mL. SD showed higher antiparasitic activity (IC50 of 21.68 ± 1.6 µM) than BNZ alone (32.47 ± 4.9 µM), likely because P407 significantly improved BNZ solubility and consequently increased the rate of its release compared to the other preparations. According to the authors, this SD could be of great help in the treatment of Chagas disease, as it would allow for more effective treatment with fewer adverse effects.

In a study carried out by Eloy et al. [29], a SD has been developed with ursolic acid, a compound that helps to prevent mass loss and muscle atrophy, inhibits fat accumulation and possesses trypanocidal activity, acting as an important ally in treatment of Chagas disease. SD was prepared by the solvent method using ursolic acid, polyoxylglyceride (Gelucire 50/13) as carrier and silica dioxide as drying agent. In in vitro tests against the trypomastigote form obtained through infection of Swiss albino mice by *T. cruzi* (strain Y), SD showed an IC50 value (219.2 µM) 44.7% lower than that of the physical mixture. These data suggest that the surfactant interacted intermolecularly with ursolic acid in the SD, enhancing its trypanocidal activity.

#### 3.2.2. Antischistosomal Activity of Solid Dispersions

Praziquantel (PZQ) is a drug widely used in the treatment of parasitic diseases, especially schistosomiasis, popularly known as “water belly” in Brazil, which is caused by contact of the parasite *Schistosoma mansoni* with the skin of the individual. Despite being the only drug in the treatment of flattened worms, PZQ has several limitations due to its low water solubility, which implies the need to administer it in high doses to patients [30,31].

Solid dispersions were developed by Perissutti et al. [30] to increase the solubility and bioavailability of this drug and then reduce its doses. SDs with PZQ were prepared using different carriers; however, the best results were obtained using PVP (Kollidon K30), PVP/VA, Kollidon-CL-M and sodium starch glycolate as polymers. Adult schistosomes were removed from female mice infected by the infectious form of *S. mansoni* (cercaria) and incubated. A ratio of 1:1 (*w*/*w*) Kollidon-CL-M:PZQ showed better performance in terms of solubility (642.54 mg/mL) and IC50 value (0.102 µg/mL) than pure PZQ (solubility of 140.30 mg/mL and IC50 of 0.165 µg/mL). According to the authors, PZQ-loaded SDs with Kollidon-CL-M could be a valid candidate for the treatment of schistosomiasis.

In another study performed by Albertini et al. [31] with the aim to improve the antiparasitic activity of PZQ, a SD with PVP K30 was prepared by the mechanochemical activation process and spray congealing technique. For in vitro testing of SDs, newly transformed schistosomula, obtained by mechanical transformation from *S. mansoni* and adult forms of the parasite were used. Adult *S. mansoni* individuals were removed from the hepatic portal and mesenteric veins of infected mice. The IC50 values of the physical mixture, pure PZQ and SD against schistosomula were 3.16, 2.58 and 2.4 µg/mL, respectively, while those of physical mixture and SD against adult parasites were 0.48 and <0.23 µg/mL. These results suggest that the SD contributed to the increase in PZQ solubility thanks to a 70% reduction of its crystallinity, thus reducing its dosage to a level suitable even for pediatric use.

#### 3.2.3. Antimalarial Activity of Solid Dispersions

Malaria is an ancient disease dating back to the fifth century BC, but today it is one of the main public health problems in the world. It is estimated to affect around 200 million people worldwide, particularly individuals living in subtropical and tropical areas, causing 500,000 deaths each year [32]. It is caused by parasites belonging to the genus *Plasmodium*, of which the most aggressive species is *P. falciparum* that multiplies in the bloodstream, whereas *P. vivax* is milder affecting less than 1% of red blood cells and hardly leads individuals to death [33].

Artemether is a drug used in combination with lumefantrine to treat malaria. In the study proposed by Fule et al. [34], SDs were developed to improve the dissolution rate of this drug, as well as to increase its solubility in water. Solid dispersions were prepared by hot-melt extrusion using a combination of several polymers, such as Soluplus, PEG 400, Lutrol F127 and Lutrol F68 and sodium lauryl sulphate as surfactant. In in vitro studies, 15 artemether dispersions formulations were tested for antimalarial activity against the protozoan *P. falciparum* 3D7 (chloroquine-sensitive cell line) in schizont form and against ITG cells (chloroquine-resistant cell line). On average, the IC50 values of SDs, ranging from 0.054 to 0.081 ng/mL, were 39 times less than that of pure artemether and about 70 times less than that of chloroquine, the standard antimalarial drug. Among the formulations, the one formed by 1.0:3.8:0.2 drug, Soluplus and PEG 400, was the most promising with an IC50 value of 0.054 ng/mL.

Lumefantrine is a medicine used in association with artemether in a 6:1 ratio to treat malaria caused by *P. vivax* strains, either sensitive or resistant to chloroquine, as well as cerebral malaria caused by *P. falciparum*. However, it has low oral bioavailability due to its low aqueous solubility, which also affects its dissolution rate. To circumvent this problem and enhance its therapeutic potential, Fule et al. [35] prepared solid dispersions of lumefantrine by hot-melt extrusion using Soluplus, Kollidon VA64 and Plasdone S630 as polymers, which were tested in vitro for their antiparasitic activity against *P. falciparum* 3D7 and ITG cells. SDs showed IC50 values of 0.084–0.213 ng/mL, while pure lumefantrine and chloroquine showed values of 18.2 and 3.8 ng/mL, respectively, which demonstrates the remarkable increase in the antiparasitic activity of this drug induced by the SDs environment.

From the studies cited in this section aimed at antiparasitic activity, it can be concluded that SDs are a valid alternative in the treatment of some parasitic diseases such as Chagas disease, schistosomiasis and malaria. Solid dispersions containing drugs already used for these diseases have in fact shown an increase in their solubility and dissolution rate, as well as a consequent improvement in the activity against parasites, showing IC50 values lower than those of pure drugs. It should also be remembered that a dose reduction can reduce the incidence of toxicity during pharmacotherapy.

### 3.3. Antimicrobial Activity of Solid Dispersions

In a study conducted by Crucitti et al. [36], the antimicrobial activity of SDs of abietic acid and chitosan prepared in different drug:polymer proportions was checked against *Staphylococcus epidermidis*. Minimum inhibitory concentrations (MICs) of abietic acid, chitosan, 1:1 drug:chitosan mixture and SD containing this mixture were 0.8, 0.5, 1.0 and 0.25 mg/mL, respectively. According to the authors, not only the amorphous state of the drug in SD but also the synergistic effect between the two components may have improved the antimicrobial activity of the bioactive compound.

The low solubility of griseofulvin, which is one of the antimicrobials generally used in the treatment of dermatophytosis caused by the fungus *Trichophyton rubrum*, makes oral administration of its formulation particularly difficult, requiring high doses to reach therapeutic levels. To get around this problem, Al-Obaidi et al. [37] have successfully increased the solubility of this drug developing forms of solvates, also known as pseudopolymorphs, consisting of drug crystals with solvents, stabilized within polymeric matrices. To obtain solid dispersions, different solvated (chloroform) and unsolvated (methanol and acetone) materials were used, including microcrystalline cellulose, PVP K30 and hydroxypropyl methylcellulose acetate succinate (HPMCAS). Griseofulvin solubility in the HPMCAS-based SD (200 μg/mL) was almost seven-fold that of the pure drug. The antifungal potential of griseofulvin and this SD was tested in vitro against *T. rubrum* NCPF 935. Biofilm production and fungal proteolytic activity were significantly reduced in the presence of SDs incorporating either solvated or unsolvated materials when compared to the control. These results demonstrated the role of polymeric dispersion in increasing the antifungal activity of this drug.

Gatifloxacin is an antibiotic used to treat various eye infections such as conjunctivitis, keratitis and keratoconjunctivitis caused by bacteria [38]. The low solubility of this drug promoted the preparation of polymeric micelles incorporating it into Pluronic F127 as polymer by the solvent diffusion method [39]. Among the dispersions obtained, the one prepared at a 0.25:2.52 (*w*/*w*) drug:polymer ratio showed the best results in terms of particle size, stability and solubility. In the agar diffusion test conducted using *Staphylococcus aureus* as the target microorganism, this dispersion yielded an inhibition zone diameter (10.33 ± 0.57 mm) 18.3% larger than that produced by commercial gatifloxacin.

Among the numerous attempts to exploit the biological potential of CM, the results of studies aimed at the preparation of CM-containing SDs with antibacterial and pesticidal activity are summarized here. Hernandez-Patlan et al. [40] prepared CM-SDs using boric acid and PVP by the solvent evaporation method. In in vitro tests, 1% boric acid, 1% SD containing 10:90 (*w*/*w*) CM:PVP and a 50:50 (*w*/*w*) mixture of them were tested against *Salmonella enteritidis* in media simulating culture, intestine and proventriculus. Boric acid led to log_10_ CFU/mL values of 7.78 ± 00 in culture, 3.81 ± 0.05 in proventriculus and even 0.00 ± 0.00 in intestine, while 1% CM:PVP of 7.73 ± 00, 3.91 ± 0.08 and 5.00 ± 0.55 and their mixture of 7.65 ± 00, 3.71 ± 0.05 and 3.94 ± 0.23, respectively. In summary, the promising results shown by the SD containing CM and PVP were even improved in the presence of boric acid, thanks to its ability to penetrate the cell membrane of the target microorganism.

Solid dispersions were also prepared to enhance the efficacy of CM in antimicrobial photodynamic therapy. Bearing in mind that CM as a photosensitizer has low solubility and phototoxicity limited to Gram-negative bacteria, Hegge et al. [41] prepared SDs using concentrated solutions of hydroxypropyl-β-cyclodextrin/CM by solvent evaporation and then HPMC by lyophilization. The antibacterial activity was tested against *Escherichia coli* (ATCC 25922) (1 × 106 CFU/mL) at concentrations of 10 and 25 µM of CM in SDs exposed to irradiation. The dispersion that showed the best results in dissolution tests was then used in in vitro tests. Since it displayed phototoxicity at both concentrations (10 and 25 µM) at a radiation dose of 5 J/cm^2^, one can infer that the dispersion increased the bactericidal power of CM against *E. coli*, so it could play an important role in photodynamic therapy.

Another study was conducted by Alves et al. [42] with the aim of developing a formulation with antimicrobial activity using CM and silver nanoparticles, as both have activity against bacteria, fungi and viruses. For this purpose, CM-containing SDs were prepared by co-precipitation using CM and P407 at 1:1, 1:2 and 2:1 (*w*/*w*) ratios and then silver nanoparticles were added. Nanoparticles were synthesized and stabilized using polyvinyl alcohol and PVP as polymers and sodium citrate. The antibacterial activity was determined in in vitro tests and expressed as MIC against *E. coli* ATCC 25922, *Pseudomonas aeruginosa* ATCC 9721 and *S. aureus* ATCC 10390. The SD composed of P407/CM/nanoparticles/PVP showed better MIC values against *P. aeruginosa* (50 μg/mL) and *E. coli* (25 μg/mL) compared to those reported in the literature (175 and 163 μg/mL, respectively) [43]. All formulations were less effective against *S. aureus* (Gram-positive) and the best performance was observed for the P407/CM/nanoparticles/citrate system. The different SD activity against Gram-positive and Gram-negative bacteria was probably due to the thinner cell wall of the latter, which may have facilitated SD internalization and cell wall disruption.

In this context, DSs with drugs and antimicrobials showed good results including increased solubility, low MIC values and reduction of bacterial colonies. Although there are currently many antimicrobial drug options on the market, they are still being used indiscriminately leading to the emergence of resistant bacteria as well as other adverse effects. Therefore, SDs not only can be an option to improve their solubility, but also require lower dosages, making antimicrobial treatment safer and more effective.

### 3.4. Antioxidant Activity of Solid Dispersions

Quercetin is a flavonoid that has, among other activities, the ability to capture free radicals. In the study by Costa et al. [44], a solid dispersion of quercetin with PVP K25 was developed with the aim of increasing its solubility and antioxidant activity determined by the 2,2-diphenyl-1-picryl-hydrazyl-hydrate (DPPH) radical assay and expressed as IC50. IC50 values of 0.61 ± 0.03 and 1.00 ± 0.02 µg/mL were obtained for SD dissolved in water and for quercetin dissolved in methanol, respectively, which demonstrates that SD enhanced quercetin antioxidant activity.

The SDs prepared by Crucitti et al. [36] using 1:1 abietic acid:chitosan already discussed in the previous section were also tested for their antioxidant activity with the same method. As expected, the antioxidant activity of chitosan (IC50 of approximately 11 mg/mL) was much lower than that of abietic acid (IC50 of 1.65 mg/mL), but the one of the 1:1 AB/chitosan SD was even stronger (IC50 of 0.61 mg/mL). There are at least two factors that can explain the particularly high antioxidant activity of SDs. First, the amorphous state may have influenced the solubility and availability of the drug compared to its crystalline nature. Secondly, the natural tendency of abietic acid to react with oxygen forming peroxides may have been reduced by its stabilization due to interaction with chitosan.

The previously mentioned CM-SD prepared by Sá et al. [16] using P407 was investigated for possible effect on the activities of acetylcholinesterase (AChE), butyrylcholinesterase (BChE), glutathione S-transferase (GST) and two isoforms (A and B) of monoamine oxidase (MAO). The SD was able to inhibit the activities of AChE, BChE and GST in aqueous medium, which suggests that the hydrophobic nature of CM was overcome by the SD amorphous state. In addition, it inhibited both MAO isoforms at a concentration of 100 μM, while pure CM had no effect at the same concentration, suggesting that CM interaction with P407 in SD was responsible for enzyme inhibition. It is known that the reaction catalyzed by MAO produces hydrogen peroxide, a source of hydroxyl radicals, so MAO inhibitors, as well as this SD, may be usefully exploited to treat pathologies associated with oxidative stress.

Coenzyme Q10 has two main functions, namely the transport of mitochondrial electrons to form phosphate for muscle contraction and the antioxidant activity of its reduced form (ubiquinol). However, its bioavailability is low and variable because of its poor solubility and high molecular weight. A study performed by Ge et al. [45] aimed to investigate the therapeutic potential of water-soluble nanoscale Coenzyme Q10-SDs with mannitol prepared by high-pressure homogenization. To assess the antioxidant activity, intracellular reactive oxygen species (ROS) were detected using myocardial cells divided into three groups treated with SDs with different particle sizes, namely 408, 300 and 146 nm and a control. Myocardial cells were subjected to hypoxia/reoxygenation to detect the free-radical scavenging ability of substances. In vitro tests showed that the ROS level in the control group increased by 96%, while those in the three other groups were reduced by 48.3–87.3%, which proved the strong antioxidant activity of SDs.

To increase water solubility of usnic acid, a secondary metabolite of lichen with antioxidant activity, Fitriani et al. [46] developed SDs with PVP K30 by spray-drying and freeze-drying and determined their antioxidant activity by the DPPH assay. Since the drug and the developed systems showed IC50 values in the range 50–100 ppm, they can be classified as moderate antioxidants, with the lowest value (63.87 µg/mL) detected for the SD prepared by freeze-drying in the proportion 1:2 (usnic acid:polymer).

Luteolin is a compound belonging to the class of biflavonoids, which is found in several plants and has a rich arsenal of therapeutic activities; however, its low solubility and instability in water pose a challenge for its use in pharmaceutical formulations. Therefore, a study promoted the production of luteolin-containing SDs by solvent evaporation, microwave irradiation or fusion using PEG 4000 as polymer [47]. After dissolving the drug (control) and the three different SDs in dimethyl sulfoxide up to a concentration of 5 mg/mL, their antioxidant activity was assessed by the DPPH assay. The antioxidant activity of the drug was 61.12 ± 4.11% and those of the above SDs 61.12 ± 4.11, 94.14 ± 6.11, 88.55 ± 3.98 and 76.23 ± 5.12%, respectively, suggesting that this property increased due to improved solubilization of luteolin in the presence of PEG 4000 as carrier.

In summary, the antioxidant activity of SDs was quantified by different methods including the activities of some enzymes, but with a certain prevalence of the DPPH assay. In the above studies aimed at therapeutic purposes, the best results have always been obtained with SDs, probably thanks to the increase in the solubility of drugs in the administration systems.

### 3.5. Anti-Inflammatory Activity of Solid Dispersions

α,β-Amyrin (ABAM) is a natural blend of pentacyclic triterpenes that has shown numerous pharmacological properties, including the anti-inflammatory activity [7]. To increase this activity, ABAM-SDs, prepared using PVP K30, PEG 6000 and HPMC by physical mixture, kneading and rotary evaporation, were submitted to the in vitro anti-inflammatory test based on nitric oxide (NO) production by lipopolysaccharide (LPS)-stimulated macrophages J774. The results showed that ABAM inhibited NO production at concentration of 20 µg/mL, although the effect of SDs was stronger than those of the control and the drug alone. Specifically, the ABAM-PVP SD prepared by the kneading method had the greatest effect with 62.49% inhibition, while ABAM showed a value of 44.55%. According to the authors, SDs prepared with hydrophilic polymers were effective in increasing the solubility of ABAM and allowed to enhance its anti-inflammatory activity.

In the cited study of Sá et al. [16], CM-SD was tested for anti-inflammatory activity via inhibition of NO production by LPS-stimulated murine macrophages (RAW 264.7). The SD was able to improve the affinity of water for CM which, in turn, exerted cytotoxic effects in the aqueous medium, with an IC50 value > 400 μg/mL. This result suggests possible SD administration in vivo, avoiding the use of toxic solvents to increase its clinical efficacy.

These studies as a whole demonstrated that the use of SDs containing low water-soluble active ingredients, both natural and synthetic, prepared with different methods, is an effective means to increase their anti-inflammatory activity, which can be evaluated in vitro by quantifying the inhibition of NO production by LPS-stimulated macrophages.

### 3.6. Cytoprotective Activity on Liver Cells

SDs of CM prepared using HPMC as the polymer have been reported by Shin et al. [48], who studied their possible hepatoprotective action in HepG2 cells treated with tert-butyl hydroperoxide (t-BHP) as a hepatotoxic agent. The survival rate of cells, which in the presence of t-BHP was about 40%, significantly increased after treatment with DSs-CM at concentrations of 10, 20 and 40 µg/mL compared to the treatment with CM alone at equivalent concentrations (2, 4 and 8 µg/mL, respectively). Additionally, the extracellular activity of lactate dehydrogenase, an enzyme released by HepG2 cells under the stress conditions induced by t-BHP, was reduced by treatment with CM-containing SDs more than by that with CM alone, indicating a stronger cytoprotective activity. Treatment with SDs containing 10 and 20 µg/mL CM was also able to inhibit, more effectively than CM alone, the activation of caspase-7, -8 and poly (ADP-ribose) polymerase, responsible for t-BHP-induced apoptosis in HepG2 cells.

In summary, given the hepatoprotective activity of CM-containing SDs, their use can be considered a valid alternative to other therapies, considering that the administration of drugs for long periods and in large doses can cause liver damage, especially in elderly patients. In this sense, further in vitro and in vivo studies on DSs of CM and other bioactive compounds are needed, given the hepatoprotective potential associated with this drug delivery system.

**Table 1 pharmaceutics-12-00933-t001:** In vitro studies on solid dispersions.

Carrier Type	Substance	Cell Type	Activity	Improved Characteristics	Reference
OHPP	Niclosamide	PC-3, HeLa, A549	Anticancer	SDs showed higher cytotoxicity to target cells (lower IC50) than the niclosamide solution.	[11]
OHPP	Paclitaxel	PC-3, HeLa, A549	Anticancer	SDs showed significantly higher cytotoxicity to target cells (lower IC50) than the paclitaxel solution.	[13]
PVP/VA TPGS	Paclitaxel	BT-474, MCF-7, SK-BR-3	Anticancer	SDs showed higher cytotoxicity against cancer cells compared to the pure drug.	[14]
Brij^®^L4	Chrysin	HT29	Anticancer	The higher solubility of chrysin in SDs compared to water solution increased cytotoxicity.	[15]
Poloxamer 407	Curcumin (CM)	NCIH460, HeLa, HepG2, MCF-7 and PLP2	Anticancer	SDs showed cytotoxicity against all tumor cell lines tested, but no toxic effects on non-tumor cells.	[16]
		AChE, BChE, GST, MAO A-B	Enzyme inhibitory /Antioxidant	SD was able to inhibit the activities of AChE, BChE and GST in aqueous medium.	
		LPS-stimulated murine macrophages (RAW 264.7)	Anti-inflammatory	IC50 (inhibitory concentration of 50% NO production by macrophages) > 400 μg/mL.	
PVP K30	Zn(II)-curcumin complex	HepG2, SK-HEP1	Anticancer	SD of Zn(II)-curcumin complex had a potent anticancer effect.	[17]
HPMC, PVP K30, PEG 6000	Telaprevir	HepG2	Anticancer	The antitumor activity was dose dependent and even with the addition of the polymer the drug maintained its efficacy.	[18]
Soluplus^®^	Angelica gigas Nakai	HeLa, HEK 293	Anticancer	SD at the concentration of 200 μg/mL showed a significant decrease (to only 17.37%) in cell viability. There was no toxicity to normal cells.	[20]
Eudragit S-100	Berberine hydrochloride (HB)	SW480, HCT116, Caco-2	Anticancer	The release of HB from SDs was effective and cell viability was reduced in a dose and time dependent manner.	[21]
PVP K30	IIIM-290	Ehrlich ascites carcinoma cells	Cytotoxic	Despite the reduced amount of IIIM-290 in SD, the IC50 value of SD was lower than that of IIIM-290 alone.	[24]
Poloxamer 407	Benznidazole	Epimastigotes of *Trypanosoma cruzi*	Antichagasic	SDs enhanced drug solubility, release kinetics and parasitic activity	[25]
Low-substituted HPC	Benznidazole	Epimastigotes and intracellular amastigotes of *T.* *cruzi* (CL-B5)	Antichagasic	SDs had higher antiparasitic activity against amastigotes than epimastigotes.	[28]
Gelucire 50/13	Ursolic acid	Trypomastigotes of *T. cruzi* Y	Antichagasic	Increased antiparasitic activity.	[29]
PVP K30,PVP/VA, Kollidon-CL-M, sodium starch glycolate	Praziquantel	Adult Schistosomes of *Schistosoma mansoni*	Antischistosomal	Increased solubility, better bioavailability and stronger antiparasitic activity.	[30]
PVP K30	Praziquantel	Newly transformed schistosomula of *S. mansoni* and adults	Antischistosomal	Increased solubility, reduced dosage especially for children and increased antiparasitic activity.	[31]
Soluplus, PEG 400, Lutrol F127 and Lutrol F68	Artemether	Schizonts of *Plasmodium falciparum* 3D7	Antimalarial	Increased dissolution rate, amorphous form, increased solubility and, mainly, increased antimalarial activity.	[34]
Soluplus, Kollidon VA64, Plasdone S630	Lumefantrine	ITG cells	Antimalarial	Increased antiparasitic activity.	[35]
Chitosan	Abietic acid	*Staphylococcus epidermidis*	Antimicrobial	SD exhibited better MIC values against *S. epidermidis* than chitosan and abietic acid alone.	[36]
		DPPH radical scavenging	Antioxidant	SD had higher antioxidant power (IC50 of 0.61 mg/mL) than abietic acid alone (IC50 of 11 mg/mL).	
PVP K30 and HPMCAS	Griseofulvin	Dermatophytes of *Trichophyton rubrum* NCPF 935	Antimicrobial	SDs significantly reduced biofilm formation when compared to the control.	[37]
Pluronic F127	Gatifloxacin	*Staphylococcus aureus*	Antimicrobial	The gatifloxacin/Pluronic F127 system exhibited antimicrobial efficacy when compared to commercialized eye drops.	[39]
PVP K30	Curcumin	*Salmonella enteritidis*	Antimicrobial	SD had a strong antimicrobial effect on *S. enteritidis*, while CM alone did not show antimicrobial activity in vitro.	[40]
HPMC	Curcumin	*Escherichia coli*	Antimicrobial	SD used to prepare phototoxic supersaturated solutions showed significant bactericidal activity against *E. coli*.	[41]
Polaxamer 407	Curcumin	*E. coli, Pseudomonas aeruginosa* and *S. aureus*	Antimicrobial	The association between SD and silver nanoparticles increased CM antimicrobial and antioxidant activities.	[42]
		DPPH radical scavenging	Antioxidant		
PVP K25	Quercetin	DPPH radical scavenging	Antioxidant	Increased quercetin antioxidant activity in SD (0.61 ± 0.03 ≤ IC50 ≤ 1.00 ± 0.02 μg/mL).	[44]
Mannitol	Coenzyme Q10	Intracellular ROS level	Antioxidant	The SD with the smallest particle size showed the greatest absorption of UVB radiation as well as the highest antioxidant activity in vitro.	[45]
PVP K30	Usnic acid	DPPH radical scavenging	Antioxidant	Increased usnic acid solubility and antioxidant activity.	[46]
PEG 4000	Luteolin	DPPH	Antioxidant	Polymers increased luteolin solubility and antioxidant activity.	[47]
PVP K30, PEG 6000 and HPMC	α,β-Amyrin	LPS-stimulated macrophages J774	Anti-inflammatory	SDs enhanced the anti-inflammatory activity of α,β-amyrin.	[7]
HPMC	Curcumin	HepG2	Cytoprotective	SDs showed better cytoprotective activity than pure CM and inhibited cell death induced by *t*-BHP.	[48]

OHPP, Octenylsuccinate hydroxypropyl phytoglycogen; IC50, Half inhibitory concentration; PVP/VA, Polyvinylpyrrolidone/vinyl acetate; TPGS, D-α-toco-pheryl polyethylene glycol-1000-succinate; PVP, Polyvinylpyrrolidone; SD, Solid dispersion; AChE: Acetylcholinesterase; BChE, Butyrylcholinesterase; CM, Curcumin; GST, Glutathione S-transferase; MAO, Monoamine oxidase; LPS, Lipopolysaccharide; NO, Nitric oxide; PEG, Polyethylene glycol; HPMC, Hydroxypropyl methylcellulose; ITG (chloroquine-resistant cell line); DPPH, 2,2-diphenyl-1-picryl-hydrazyl-hydrate; HPMCAS, Hydroxypropyl methylcellulose acetate succinate; MIC, Minimum inhibitory concentration; ROS, Reactive oxygen species; t-BHP, *tert*-Butylhydroperoxide.

## 4. In Vivo Studies on Solid Dispersions in Polymeric Matrices

As already mentioned, solid dispersions (SDs) have been used as a strategy in pharmaceutical technology to circumvent some limitations presented by drugs and new bioactive compounds such as low solubility and bioavailability. In this sense, this section addresses in vivo studies on SDs with different biological activities, as shown in Figure 3 and quantitatively expressed in Figure 4. The main information about these studies is summarized in Table 2.

### 4.1. Anticancer Activity of Solid Dispersions

As previously mentioned, the low solubility of many active compounds complicates oral formulations requiring high doses. Therefore, SDs have been prepared to increase their solubility and, consequently, their therapeutic effects. Most SDs that were successful in in vitro experiments described previously in Section 3 were also assessed in vivo assays, whose results are summarized and critically discussed here together with those of other studies.

The IIIM-290-containing SDs prepared by Kumar et al. [24] with the highest solubility were assessed for in vivo anticancer activity on mice with Ehrlich ascites carcinoma. The animals were divided into four groups to receive treatment and two controls. Group I received only IIIM-290 (50 mg/kg), while groups II, III and IV received SD consisting of PVP K30 and drug at doses of 25, 50 and 75 mg/kg, respectively. While the group that received the drug alone showed a tumor inhibition rate of only 24.75%, the SD reduced tumor growth by 31–43% and ensured better performance even at a half dosage of drug.

To improve the solubility and permeability of 9-nitrocamptothecin, a topoisomerase I inhibitor and potent chemotherapy drug to treat pancreatic cancer, a SD was developed using Soluplus by lyophilization and its antitumor activity was in vivo assessed in mice with sarcoma 180 [49]. SD showed higher tumor growth inhibitory rate (approximately 80%) than the drug alone due to improved oral bioavailability.

(−)-Oleocanthal, a phenolic compound derived from olive oil with anti-inflammatory, antitumor and anti-Alzheimer activities, exerts an irritating action in the oropharyngeal region that limits its use in oral formulations. To improve its taste and dissolution, Qusa et al. [50] developed by the hot-melting method a SD containing 1:7 oleocanthal:xylitol, whose antitumor activity was tested in vivo against human triple-negative breast cancer cells (MDA-MB-231) xenografted in mice. SD administration resulted in a two-day delay and 49% tumor growth rate when compared with the group treated only with placebo. Additionally, in the recurrence test the SD-treated group showed recurrence-free survival in two days, with only two of the five animals suffering from recurrent but smaller tumors, while in the control group a 100% tumor reappearance rate was observed.

Stimulated by the promising results obtained in vitro, Wu et al. [17] also tested in vivo the antitumor effect of Zn(II)-curcumin (CM) SDs with PVP K30. Hepatic xenografts of H22 and HepG2 cells were performed in male BALB/c mice and female B-NDG mice, respectively. BALB/c mice were divided into three groups, in which each received a dose of PVP K30 (700 mg/kg), or CM-SD or Zn-CM-SD (equivalent doses of 100 mg/kg CM and ZnCM), while B-NDG mice were treated with the same formulations except CM-SD at the same doses. Using H22 cells, while the untreated group had tumors weighing over 3 g and volume of 2000 mm^3^, tumors in animals treated with Zn-CM-SD weighed less than 2 g and had a volume of less than 1000 mm^3^. In addition, greater extension of necrotic tissue but no reduction in animal body weight was reported.

*Selaginella doederleinii* Hieron is a widely distributed medicinal herb in southern China that has traditionally been used as a folk medicine for cancer, especially lung cancer. However, its total biflavonoid extract (TBESD) has low water solubility, gastrointestinal permeability and oral absorption that limit its use in therapeutic formulations [51,52].

To overcome these difficulties, Chen et al. [53] prepared a solid dispersion of TBESD with the polymer PVP K30 (TBESD-SD) by the solvent evaporation method. To assess the in vivo antitumor effect, mice with xenograft of A549 cells derived from human alveolar basal epithelial cell adenocarcinoma were treated with doses of 200 mg/kg TBSED and TBESD-SD and 2 mg/kg doxorubicin. Tumor growth inhibition rates in the TBESD, TBESD-SD and pure drug groups were 29.48, 46 and 58.44%, respectively and the respective reduction rates of tumor angiogenesis 24.30, 52.20 and 59.91%. However, the drug group, unlike the other two, showed significant body weight loss. Therefore, the dispersion developed in this study proved to be a promising formulation for the treatment of tumors.

In summary, it can be concluded, despite the limited number of studies concerning the in vivo antitumor activity, that the use of SDs allowed a higher rate of tumor inhibition than the isolated compounds, also corroborating the results obtained in vitro in other studies.

### 4.2. Antiparasitic Activity of Solid Dispersions

The Benznidazole (BNZ)-containing SDs developed by Fonseca-Berzal et al. [28] using sodium deoxycholate (NaDC) as surfactant and low-substituted hydroxypropyl cellulose (HPC) as polymer were also tested in vivo for trypanocidal activity. For this purpose, parasites of *Trypanosoma cruzi* (strain Y) isolated from human individuals in the acute phase of the disease were used in their trypomastigote form to infect mice, which were administered at a 25 mg/kg/day oral dosage. The percentage reductions in the area under the parasitic curve were 91.93, 96.65 and 95.27% administering SD with 1:3 BNZ:NaDC, SD with 1:3 BNZ-HPC and BNZ alone, with the best anti-interference performance obtained for the second formulation, while the survival rate of in the control group was only 44.44%. None of the mice treated with SDs or BNZ alone died, which indicates no toxic effects at the established dosage. Although these SDs have shown great potential in in vivo tests, further studies are needed because BNZ has a reduced pharmacological activity in the chronic phase of Chagas disease and has several side effects.

### 4.3. Antimicrobial Activity of Solid Dispersions

In the study developed by Hernandez-Patlan et al. [40], the antibacterial activity of CM-loaded SDs prepared using PVP K30 and boric acid was evaluated against *Salmonella enteritidis*. In vivo tests were performed to determine the effects of (a) SD with 10/90 (*w*/*w*) CM/PVP, (b) boric acid and (c) 50/50 (*w*/*w*) CM/PVP plus boric acid administered orally to broiler chickens treated with 10^7^ CFU/mL *S. enteritidis* each. While boric acid had no significant effect, the CM/PVP plus boric acid triad significantly reduced the number of colonies in the intestine of animals compared to CM/PVP, revealing a synergistic effect. Since such a reduction occurred only in the above CM/PVP proportion in SD, it can be concluded that this effect was the result of the increase in CM solubility. Although this in vivo antimicrobial activity of CM-SDs would allow for equivalent or even greater therapeutic effects than the drug alone in lower doses, further studies involving other species of target microorganisms and animals are needed.

### 4.4. Antioxidant Activity of Solid Dispersions

Some studies have revealed that zinc plays an important role in maintaining redox balance as well as protecting against oxidative stress and damage induced by ethanol, a compound capable of reducing the defenses of the gastric mucosa and inducing gastric ulcers [54,55]. Taurine is an amino acid that participates in numerous physiological reactions and can exert a cytoprotective effect against lesions of the stomach mucosa [56] thanks to its antioxidant activity [57].

In this sense, SDs of the taurine-zinc complex with the PVP K30 were prepared by Yu et al. [58] in the 1:6 (*w*/*w*) proportion by spray drying. In the in vivo study, the gastric mucosa of rats with ethanol-induced lesions was examined. The treatment with SD taurine-zinc (200 mg/kg) led to a 46% increase in the superoxide dismutase (SOD) activity and a 3.9-fold increase in the level of glutathione in the gastric mucosa compared to the model group (28.40 U/mg protein and 0.81 ± 0.56 mg/g protein, respectively), which suggests a direct relationship between endogenous level of this antioxidant and SOD activity to be exploited for better therapeutic response. Unlike in vitro activity, we found in the databases only this study published on in vivo antioxidant activity of SDs, which highlights the need for further research in this field.

### 4.5. Anti-Inflammatory Activity of Solid Dispersions

In an anti-inflammatory study conducted by Gou et al. [59], a solid dispersion of triacetylated andrographolide (TA) with Kollidon (VA64) had its in vivo anti-inflammatory activity assessed in mice ulcerative colitis model through parameters such as length, weight and levels of inflammatory cytokines (IL-2, IL-6) and SOD activity in the colon. Reduced length and increased weight of the colon indicated severe edema in the model group and in the group administered with TA suspension. Significant differences were also observed in these parameters between the model/crude TA group and the groups treated with TA-SD at different doses. While high a dose (200 mg/kg/day) of TA-SD remarkably reduced colon weight, low (50 mg/kg/day), or medium (100 mg/kg/day) doses showed a therapeutic effect against ulcerative colitis comparable to that of andrographolide pills (commercial anti-inflammatory drug). In addition, a reduction in the levels of IL-2 and IL-6 cytokines was observed in both the TA-SD and pill groups compared to the model group and the crude TA group. Finally, SOD activity significantly decreased in the model group compared to the normal one, while administration of TA-SD at medium and high doses caused an increase in SOD activity comparable to that induced by andrographolide pills. These results taken together demonstrated the anti-inflammatory effect of TA-SD.

Different solid dispersions of curcumin with PVP K30, Poloxamer 188 or 2-hydroxypropyl-β-cyclodextrin (HP-β-CD) were prepared by the solvent evaporation method and their anti-inflammatory activity was tested in mouse ear edema model [60]. The histological cuts of the ears revealed a decrease in the number of inflammatory cells infiltrated when mice were submitted to treatment with curcumin and CM-PVP. The results of ELISA assay showed a stronger anti-inflammatory activity of SDs compared to pure CM, revealed by a reduction in the expression of matrix metallopeptidase 9 as well as the levels of IL-1β and IL-6 cytokines.

Still regarding curcumin, Teixeira et al. [61], with the aim of improving its solubility, prepared atomized ternary SDs using Gelucire^®^50/13-Aerosil^®^, which provided important results in terms of anti-inflammatory activity in vivo. Administration of SDs containing 40% CM in a dose of 100 mg/kg showed a greater capacity to reduce rat paw edema than indomethacin (5 mg/kg). This anti-inflammatory activity, which appeared to be dose-dependent in the 10–100 mg/kg range, was ascribed to the increase in gastrointestinal absorption of SDs observed in the study.

In another research, Chuah et al. [62] investigated the bioavailability and bio-efficacy of CM in a solid dispersion prepared using HPMC, lecithin and isomalt by hot-melt extrusion for possible application in food products. To test the bio-efficacy of CM-SD on male Sprague-Dawley rats, they used the in vivo LPS-induced lung inflammation model and a dose of 5 mg/kg in 0.25% carboxymethylcellulose as vehicle administered for 21 days by oral gavage. Administration of LPS resulted in acute inflammatory lung injury and septic shock. Because the bioavailability of CM was increased by more than 10 times in solid dispersion, its efficacy was tested at a concentration 10 times lower than that of unformulated CM. While the treatment with CM alone at 50 mg/kg for 21 days reduced the leukocyte count marginally, that with CM-SD at only 5 mg of CM/kg significantly reduced the counts of both bronchoalveolar lavage fluid leukocytes and neutrophils. Damages in the lung tissues were less pronounced in CM-SD treated rats compared to those treated with LPS or CM alone. CM-SD, but not unformulated CM, was also able to downregulate the levels of IL-1, IL-6 and IL-8 inflammatory cytokines significantly. In summary, despite a 10 times lower dose, CM-SD was responsible for greater anti-inflammatory activity than CM alone. This study, which is the first report on CM-SDs suitable for food, demonstrated that the use of CM in solid dispersion in significantly low doses does not affect its efficacy and may improve color, taste and smell of foods.

Aceclofenac, chemically [2-(2′,6′-dichlorophenyl)amino] phenylacetoxyacetic acid, is used as a non-steroidal anti-inflammatory drug in the symptomatic treatment of pain and inflammation. However, like other drugs of this class, its oral administration is associated with gastrointestinal side effects like gastric ulceration, gastrointestinal bleeding and liver and kidney trouble. For these reasons, aceclofenac is increasingly administered by topical route. In this context, Carbopol gels offer a good alternative to oil-based topical formulations [63] and crospovidone has been used as a carrier of SDs of various drugs to improve their aqueous solubility [64]. In the study performed by Jana et al. [65], an attempt was made to develop Carbopol 940 gel for topical application containing SDs of aceclofenac using crospovidone as carrier to improve the skin permeation profile of this drug. The aceclofenac-crospovidone (1:4) SD was prepared by the solvent evaporation technique and tested in vivo for anti-inflammatory activity on male Sprague-Dawley rats using carrageenan-induced rat-paw edema model. The results obtained applying 1 g of optimized gel on the skin (1 cm^2^) back of rats showed that SDs increased the rate of permeation of the drug, increased the intensity of response and gave results comparable with those of a marketed gel without leading to any skin irritation.

Another non-steroidal anti-inflammatory drug widely used in the treatment of mild-to-moderate pain and fever is ibuprofen ([2RS]-1[4-(2-methyl propyl) phenyl] propionic acid). Ofokansi et al. [66] tested in vivo the anti-inflammatory activity of ibuprofen-loaded PEG 8000 SDs prepared by the fusion method. The paw edema test was applied on Wister rats by orally administering ibuprofen-SD doses equivalent to 20 mg/kg body weight, while the reference group received 20 mg/kg of pure drug and the control one 1% (*w*/*v*) PEG 8000 suspension used as vehicle. Thirty minutes after treatment, paw edema was induced by injection of egg albumin. All SDs showed good anti-inflammatory properties being able to inhibit up to 90% of edema after 6 h versus only 77% of inhibition observed with the drug alone. This study clearly demonstrated that the use of hydrophilic carriers such as PEG 8000 can improve the drug pharmacological activity.

To understand and improve the dissolution behavior of poorly water-soluble drugs from SDs, flurbiprofen, a phenylalkanoic acid derivative belonging to the non-steroidal anti-inflammatory drugs, was used as a model [67]. SDs were prepared by the fusion method using urea and mannitol as hydrophilic carriers. Microspheres were also prepared by the solvent evaporation method using Eudragit L-100 (EL100) and Eudragit RS PO (ERS) as rate-controlling polymers. Flurbiprofen:urea-SDs and microspheres were used to develop controlled-release formulations by mixing them in different proportions. In in vivo anti-inflammatory tests, the pure drug and formulations were administered orally at a dose of 11.69 mg/kg body weight. The formulation that showed the best inhibition of rat paw edema up to 16 h was identified as a suitable product for controlled drug delivery.

Still within this context, meloxicam is a potent nonsteroidal anti-inflammatory drug belonging to the selective cyclo-oxygenase II inhibitors with the advantage of being safer in the gastrointestinal tract; however, it is poorly water-soluble and highly permeable (class II). A study by Al-Remawi et al. [68] aimed to improve meloxicam solubility and maximizing its pharmacological activity by preparing, by rotary evaporation and freeze drying, binary SDs with paracetamol, a highly water-soluble and poorly permeable drug. The anti-inflammatory effect was also investigated by measuring the changes in tail volume of edema induced by carrageenan injection. The formulation prepared by lyophilization using ethanol as a solvent and a meloxicam:paracetamol ratio of 1:10 (*w*/*w*) showed a decrease of more than 50% in the volume of carrageenan-induced tail edema compared to the physical mixture.

Chelerythrine (CHE), a quaternary benzo[c]phenanthridine alkaloid widely distributed in the Papaveraceae and Rutaceae families of plants, is a medicine officially listed in the Chinese Pharmacopoeia. Although this drug has shown a wide range of biological activities, its use in formulations is hampered by its poor water solubility. With the aim of improving CHE solubility and bioavailability, Li et al. [69] prepared solid dispersions with PVP K30 by the solvent evaporation method. Their anti-inflammatory activity was assessed in LPS-induced endotoxic shock experiments after single dose administration of dispersions (CHE-SDs) to mice in comparison with the CHE solution. CHE-SDs (10 mg/kg) significantly improved the bioavailability and anti-inflammatory activity of CHE by reducing the levels of TNF-α, IL-6 and NO in mice serum.

In summary, most in vivo studies on the anti-inflammatory activity used the paw edema test in rats, which revealed a significant increase in the biological activity of drugs in SDs compared to pure drugs, reducing lesion size and inflammatory cytokines production. These results corroborate those of tests often carried out by the same authors on solubility, bioavailability and physicochemical properties of drugs, which are directly related to the pharmacokinetic characteristics of SDs.

### 4.6. Gastro and Hepatoprotective Activity of Solid Dispersions

Among the activities and effects reported in the literature for SDs, the gastroprotective and hepatoprotective ones were also found.

Regarding the gastroprotective effect, a dispersion of the zinc taurine complex with PVP K30, already mentioned in the section on antioxidant activity, was also tested in vivo for its gastroprotective effect in experiments conducted on rats with ethanol-induced lesions in the gastric mucosa [58]. The gastric mucosa pretreated with taurine-zinc SD had an almost normal appearance, while that of the untreated group exhibited severe hemorrhagic ulcers. In addition, administration of SD reduced the ulcer index from 124.63 ± 26.72 in the untreated group to no more than 13.25 ± 4.77 in that treated with 200 mg/kg of SD. The gastroprotective effect of SDs was confirmed by a reduction in serum NO levels and a significant increase in gastric prostaglandin E2 compared to the untreated group.

Instead, Shin et al. [48] tested in vivo the hepatoprotective activity of CM-containing SDs with HPMC against tert-butyl hydroperoxide (t-BHP)-treated mice by administering 80 mg/kg of CM (positive control) or an equivalent dose of SD (400 mg/kg). Levels of alanine aminotransferase (ALT) and aspartate aminotransferase (AST), which indicate liver damage in response to t-BHP, were reduced by SD, while CM alone had no statistically significant effect. In the same study, the hepatoprotective effects of dispersion in liver tissues of mice were also evaluated. The liver section of the CM-administered group showed a slightly reduced necrotic region and decreased cellular infiltration, whereas these effects were much more marked in the SD-administered groups (200 and 400 mg/kg), with consequent improvement of the histological damage caused to the liver.

Solid dispersions of nobiletin and hydroxypropyl cellulose were also developed with the aim of improving drug solubility and bioavailability, as well as detecting possible hepatoprotective properties. Nobiletin is a polymethoxylated flavone with pharmacological activities such as anti-inflammatory, anti-apoptotic, anti-dementia, anti-oxidative and anti-tumor activities. The method used by Onoue et al. [70] to prepare nobiletin-SD was wet-milling followed by freeze drying. The evaluation of hepatoprotective activity was conducted in a rat model with acute liver damage induced by carbon tetrachloride (CCl_4_). Administration of 2 mg/kg of SD led to a significant decrease in liver damage, evidenced by 60 and 43% reductions in ALT and AST levels and to greater hepatoprotective efficacy compared to the crystalline drug.

Silymarin is an extract of *Silybum marianum* seeds, used in the treatment of liver diseases, which however has low solubility in water. SDs and spherical agglomerates of silymarin were developed by Balata and Shamrool [71] to investigate their therapeutic effects on liver and kidney diseases. SDs with PVP K30 were prepared by the solvent evaporation method in 1:2, 1:4 and 1:8 (*w*/*w*) extract:polymer proportions. In vivo liver tests were conducted on albino rats with CCl_4_-induced hepatotoxicity. The group treated with the SD with the lowest extract content significantly reduced the serum activities of ALT, AST and ALP as well as the level of total bilirubin, but increased the activity of alkaline phosphatase compared to the untreated group. In summary, this SD improved biomarker rates and had a significantly better therapeutic effect than the commercial extract. Furthermore, the results of histopathological studies confirmed the SD potential to reverse the chemically induced liver toxicity.

Another study with silymarin developed by Hwang et al. [72] aimed to evaluate a new system of SDs prepared using PVP K30 and Tween 80 by spray drying. Adhesion of the former ingredient to the outer surface of the drug led to a 650-fold increase in silymarin solubility. The hepatoprotective activity was determined on male rats and the tested substances were administered before and after exposure to CCl_4_. The level of AST, which had been increased by exposition to CCl_4_, was significantly reduced by the administration of silymarin-containing SD, while both the raw powder and commercial product did not show any appreciable hepatoprotective effect. These results agree with those of the histopathological study.

In summary, with regard to gastro and hepatoprotection, an evident increase in the biological activity of drugs was observed when they were incorporated into polymers, which reduced the level of damage more than the isolated drugs, thus demonstrating that SD have advantages that go beyond the increase in aqueous solubility and improvement of physicochemical properties of pharmacologically active molecules.

### 4.7. Antidiabetic Activity of Solid Dispersions

Diabetes mellitus is an endocrine-metabolic disorder characterized by chronic hyperglycemia, which can cause microvascular damage (retinopathy, nephropathy and neuropathy) and macrovascular damage (ischemic heart disease, stroke and peripheral vascular disease) [5,73].

Repaglinide, a hypoglycemic drug indicated for the treatment of patients with type 2 diabetes mellitus, was used by Zawar and Bari [74] to develop dispersions with Poloxamer 188 in the 1:3 (*w*/*w*) drug:polymer proportion by the conventional fusion or microwave method. In vivo experiments conducted on Wistar rats showed that the SD prepared by the microwave method reduced the serum glucose level determined by the glucose-oxidase method to 88 ± 0.73 mg/dL within 2 h, while the pure drug and the control led to glucose levels of 94 ± 2.63 and 120 ± 2.1 mg/dL, respectively. This study has shown that SDs can be an effective way to increase the solubility and bioavailability of repaglinide, leading to an enhancement of its antihyperglycemic activity and that the microwave method can be a green and effective alternative to obtain them.

Among the substances used in the treatment of non-insulin dependent diabetes mellitus is glimepiride, one of the sulfonylurea antidiabetics responsible for stimulating insulin secretion. These drugs have low solubility and high permeability and their use in the treatment of diabetes has been impaired by side effects [75]. In this sense, glimepiride-SDs were prepared by the solvent evaporation method using mixtures of Soluplus1 and PEG 4000 as polymers in different proportions and their antidiabetic activity in vivo was determined by changes in blood glucose concentrations in albino rats [5]. Glycemia was reduced by as much as 90.2–91.0% after 3 h in rats treated with the most effective SDs compared to the pure drug (38%) and commercial formulation (Amaryl1) (85%) and glucose in blood remained within the normoglycemic level for 9–12 h.

Pioglitazone is an antidiabetic drug that acts reducing glucose production in the liver and increasing insulin-dependent glucose availability. However, limitations related to its low water solubility and dissolution have led Pokharkar et al. [76] to prepare solid dispersions using different polymers such as HPMC E3, PVP K17, PVP K30, Soluplus and surfactants including Tween 80, Poloxamer 407 and Labrasol. The PVP K17-SD showed a better dissolution profile in vitro compared to pure pioglitazone, therefore it was chosen to assess hypoglycemic activity in vivo. This dispersion exerted an anti-hyperglycemic effect that resulted within 10 h in a serum glucose level (270.50 ± 4.47 mg/dL) approximately 20% lower than that observed using pure pioglitazone or a commercial drug, corresponding to a 33.85% reduction in concentration versus about 17%. The results of this study showed that SD was effective in improving the physicochemical properties of pioglitazone and may contribute to medical treatments as a hypoglycemic drug.

### 4.8. Antinociceptive Activity of Solid Dispersions

Neuropathic pain is a state of chronic pain caused by lesions or diseases of the somatosensory nervous system [77], for the treatment of which different drugs are used, which; however, have shown inconsistent results. To develop alternative therapies, among the pharmacological activities attributed to hecogenin acetate, the analgesic one has proven interesting in the treatment of chronic pain [78]. In this context Moreira et al. [77] developed solid dispersions with hydrophilic polymers, namely PVP K30, PEG 6000, HPMC, Soluplus^®^ and PVP/VA 64, to increase the solubility of this drug and improve its dissolution profile. In this study, the anti-hyperalgesic activity of SDs was evaluated in vivo in mice selected as a neuropathic pain model. To assess the hyperalgesia of formulations, after induction of damage in mice by crushing the sciatic nerve, hecogenin acetate or its SD prepared with HPMC were administered and the parameters of mechanical and thermal hyperalgesia as well as grip strength were evaluated. The thermal hyperalgesia test on groups treated with SD showed that the incorporation of the drug in the HPMC matrix significantly improved its antinociceptive activity, while mice treated with hecogenin acetate and SD-HPMC only showed a significant effect on nociceptive behavior after mechanical and thermal stimulus testing.

**Table 2 pharmaceutics-12-00933-t002:** In vivo studies on solid dispersions.

Carrier Type	Substance	Animal	Dose	Activity	Improved Characteristics	References
PVP K30	IIIM-290	Swiss male mice (18–23 g)	25, 50 and 75 mg/kg	Anticancer	SD was able to reduce the IIIM-290 dose by at least 1.5-fold thanks to its efficiency in Ehrlich solid tumor model.	[24]
Soluplus^®^	9-Nitrocamptothecin	Male Sprague-Dawley rats (20 ± 2 g)	4 mg/kg	Anticancer	SD showed higher tumor growth inhibitory rate than the pure compound due to improved oral bioavailability	[49]
(+)-Xylitol	(−)-Oleocanthal	Athymic nude mice	10 mg/kg	Anticancer	Treatment with SD showed prevention, lower growth rate and less recurrence of tumor.	[50]
PVP K30	Zinc(II)-curcumin complex	B-NDG, BALB/c mice	100 mg/kg	Anticancer	SD reduced tumor size and weight in animals.	[17]
PVP K30	*Selaginella doederleinii* Hieron	BALB/c mice	200 mg/kg	Anticancer	SD reduced tumor size as well as the level of tumor angiogenesis.	[53]
Low substituted HPC	Benznidazole	Female NMRI mice (25 ± 2 g)	25 mg/kg/day	Antichagasic	The best SD showed a 96.65% trypanocidal activity, expressed as percentage reduction in the area under the parasitic curve.	[28]
PVP K30	Curcumin (CM)	Male Cobb-Vantress broiler chickens	1 g/kg of feed	Antimicrobial	The synergistic effect of 0.05% CM/PVP SD with 0.05% boric acid reduced colonization of *Salmonella enteritidis* in crop and ceca-cecal tonsils.	[40]
PVP K30	Taurine-zinc complex	Female Sprague-Dawley rats (240–260 g)	100 and 200 mg/kg/day	Antioxidant	SDs protected rat gastric mucosa from ethanol-induced injury and increased SOD activity and glutathione level.	[58]
				Gastroprotective		
Kollidon (VA64)	Triacetylated andrographolide (TA)	Male Kunming mice	50, 100 and 200 mg/kg/day	Anti-inflammatory	TA-SD prepared with VA64 significantly improved the drug activity against ulcerative colitis.	[59]
PVP K30, Poloxamer 188	Curcumin	Female CD-1 mice	100 mg/kg oral doses	Anti-inflammatory	CM-SD prepared with PVP decreased matrix metallo-peptidase 9 expression and levels of IL-1β and IL-6 cytokines.	[60]
Gelucire^®^50/13-Aerosil^®^	Curcumin	Rat	10 to 100 mg/kg	Anti-inflammatory	A CM-SD dose of 100 mg/kg was more effective than 5 mg/kg indomethacin in reducing edema.	[61]
HPMC, lecithin and isomalt	Curcumin	Male Sprague-Dawley rats	5 mg/kg	Anti-inflammatory	A CM-SD dose of 5 mg/kg had greater anti-inflammatory activity than 50 mg/kg curcumin alone.	[62]
Crospovidone	Aceclofenac	Male Sprague-Dawley rats	1 g/cm^2^ (topical)	Anti-inflammatory	The enhanced drug permeation increased the intensity of the anti-inflammatory response.	[65]
PEG 8000	Ibuprofen	Wistar rats	20 mg/kg	Anti-inflammatory	All SDs showed better anti-inflammatory activity than the pure drug, allowing up to 90% edema inhibition after 6 h.	[66]
Urea and mannitol	Flurbiprofen	Rat	11.69 mg/kg	Anti-inflammatory	SD showed better inhibition of rat paw edema up to 16 h.	[67]
Paracetamol	Meloxicam	Rat	-	Anti-inflammatory	SDs reduced by more than 50% the volume of carrageenan-induced tail edema compared to the physical mixture.	[68]
PVP K30	Chelerythrine (CHE)	Mice	10 mg/kg	Anti-inflammatory	SD enhanced CHE anti-inflammatory effect by reducing the levels of TNF-α, IL-6 and NO in mice serum.	[69]
HPMC	Curcumin	Male BABL/c mice	200 and 400 mg/kg	Hepatoprotective	The best SD increased the hepatoprotective efficacy of CM.	[48]
HPC	Nobiletin	Male Sprague-Dawley rats(220 g)	2 mg of drug/kg	Hepatoprotective	SD was more effective than the crystalline drug in rats with acute liver injury.	[70]
PVP K30	Silymarin	Adult male albino rats (150–200 g)	25 mg/kg	Hepatoprotective	The best SD improved biomarker rates and had a significantly better hepatoprotective effect than the commercial extract.	[71]
PVP K30	Silymarin	Male Sprague-Dawley rats (190–210 g)	50 mg/kg	Hepatoprotective	SD improved drug solubility and hepatoprotective activity, reducing the AST levels.	[72]
Poloxamer 188	Repaglinide	Wistar rats (150–250 g)	(1 mg of drug)	Antihyperglycemic	SD obtained by the microwave method improved the drug anti-hyperglycemic activity.	[74]
Soluplus1 and PEG 4000	Glimepiride	Albino Wistar rats (200–250 g)	0.0285 mg of drug/kg	Anti-diabetic	SD reduced the glucose level in rats more than the pure drug and a commercial product.	[5]
PVP K17	Pioglitazone	Male Swiss albino mice (25–30 g)	30 mg/kg SD	Antihyperglycemic	SD reduced the mean glucose level in mice more than the pure drug and a commercial product.	[76]
HPMC	Hecogenin acetate	Male Swiss mice (28–35 g)	40 mg/kg	Antinociceptive	Both the drug alone and its SD with HPMC-reduced mechanical and thermal hyperalgesia induced by crushing of the sciatic nerve in mice.	[77]

PVP, Polyvinylpyrrolidone; SD, Solid dispersion; HPC, Hydroxypropyl cellulose; CM, Curcumin; CM/PVP SD, curcumin/polyvinylpyrrolidone solid dispersion; SOD, Superoxide dismutase; PEG, Polyethylene glycol; TNF-α, tumor necrosis factor alpha; IL, interleukin; NO, Nitric oxide; HPMC, Hydroxypropyl methylcellulose; AST, Aspartate aminotransferase.

## 5. Critical Analysis

In this review, the main studies on the biological activities of solid dispersions (SDs) have been examined to make a comparison of the results of in vitro and in vivo tests carried out in the last 10 years. In vitro studies have demonstrated the ability of SDs incorporating various types of active compounds to enhance their antitumor, antiparasitic, antimicrobial, antioxidant, anti-inflammatory, or cytoprotective activities, while additional activities, such as the gastroprotective, hepatoprotective, antidiabetic, or antinociceptive ones, have been highlighted by in vivo studies.

Most of the articles published and cited in this review, for a total of 33, refer to in vitro studies particularly addressed to the antitumor activity of active compounds (cell viability or cytotoxicity), while the in vivo ones were relatively fewer in number, equal to 26 and mostly devoted to their anti-inflammatory activity.

Although SDs have already been studied and cited in the literature, the number of studies published on them with this focus is still relatively small, considering the great potential of these formulations in pharmaceutical technology and the possibility of preparing both amorphous and crystalline ones for the most diverse applications. In this review, different processes, carriers and drugs to prepare SDs were also discussed, a variety that can be of great value in the field of pharmaceutical technology to develop new SDs for the release of a large number of different drugs.

The results of biological activity studies have shown that SDs, as a tool of drug release, do not constitute a limiting factor for the execution of in vitro and in vivo tests; indeed, they stand out as a promising system in which the active principle and the vector interact allowing, in most cases, an increase in the pharmacological potential of the former as a consequence of changes in the physicochemical properties of the constituents. Similarly, SDs can enhance the biological properties of the incorporated bioactive compounds, making them more effective against a wide variety of pathogens such as bacteria and protozoa.

As a final remark, SDs could play a role in various biological mechanisms, thus representing a safe and effective alternative for the development and improvement of drugs targeting a wide range of pharmacological treatments.

## 6. Conclusions

Solid dispersions are a technological strategy to improve the pharmacological potential of natural or synthetic bioactive molecules, thanks to an increase in their solubility and bioavailability, thus leading to possible enhancement of their biological activities. In this context, this review sought to summarize and critically examine studies conducted with this approach both in vitro and in vivo, which have provided evidence of a significant therapeutic potential that solid dispersions could offer in the context of the most diverse biological activities.

As poorly water-soluble active ingredients incorporated in solid dispersions exhibited improved physicochemical and pharmacological properties, these systems can be targeted in the pharmaceutical industry as possible therapeutic alternatives for certain diseases, with greater selectivity, safety and efficacy than the existing treatments.

## Figures and Tables

**Figure 1 pharmaceutics-12-00933-f001:**
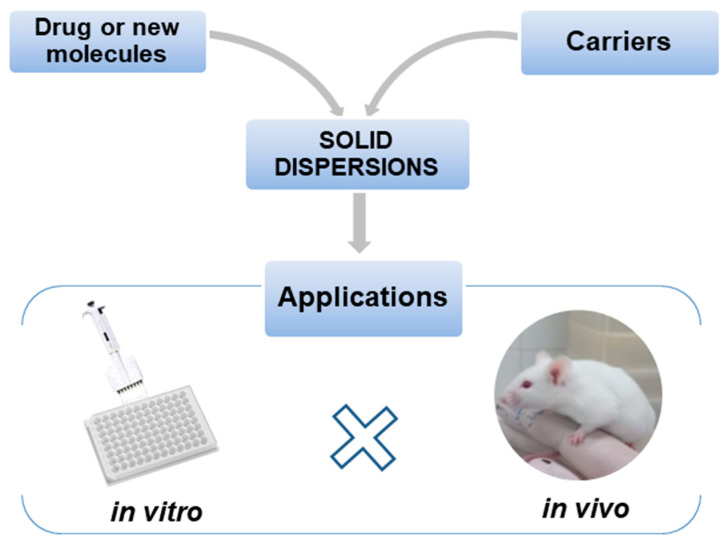
Representation of the application of solid dispersions in biological assays.

**Figure 2 pharmaceutics-12-00933-f002:**
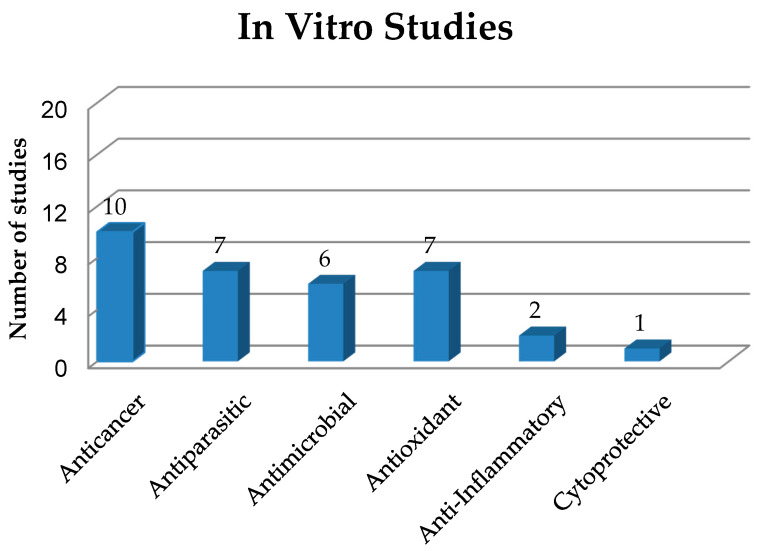
Quantification and classification of in vitro studies on solid dispersions published in the period from 2009 to 2020.

**Figure 3 pharmaceutics-12-00933-f003:**
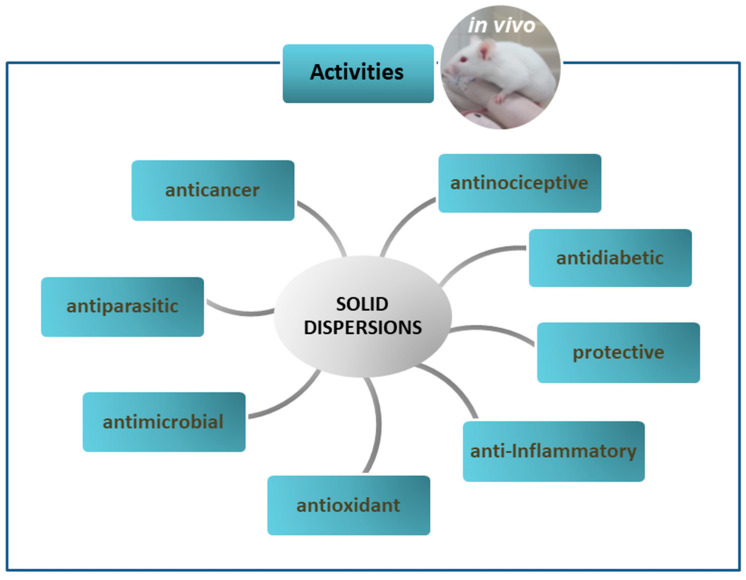
In vivo activities of solid dispersions.

**Figure 4 pharmaceutics-12-00933-f004:**
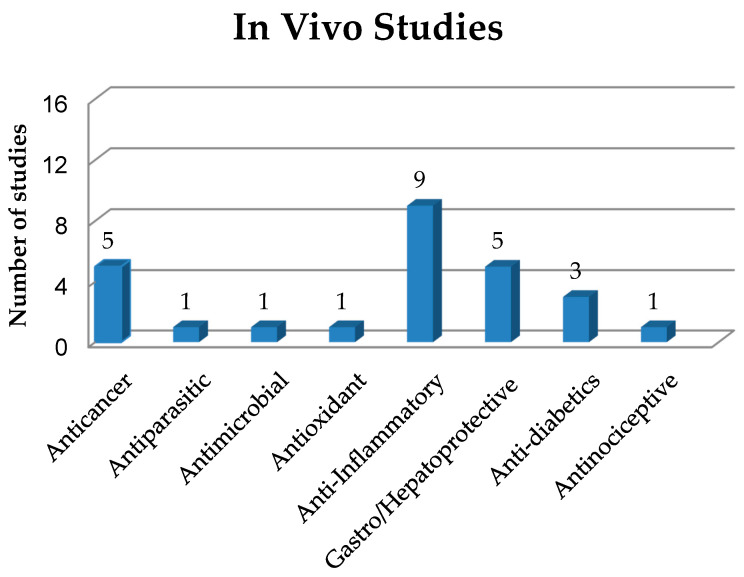
Quantification and classification of in vivo studies on solid dispersions published in the period from 2009 to 2020.

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
