# Peer review of "Therapeutic Applications of Solid Dispersions for Drugs and New Molecules: In Vitro and In Vivo Activities"

_pharmaceutics, 2020, doi:10.3390/pharmaceutics12100933_

Round 1
Reviewer 1 Report
The manuscript provides an interesting overview of the use of solid dispersions, principally the polymeric matrices, as vehicles for bioactive compounds with different biological activities, such as anticancer, antimicrobial, anti-inflammatory and so on. Both the in vitro and in vivo biological assays collected highlighted the advantages of the solid dispersion systems, due to the increase of biological potentials of the active substances and the improvement of their physicochemical properties.
The review is appropriately designed, the data are exposed clearly, and the state-of-art is well documented. Some improvements could be required, just to improve the study.
Comment on manuscript:
Line 46: I would move the reference at the end of the sentences, because it is referred both to sucrose, dextrose, and galactose neither to the high thermodynamic stability.
Line 54-57: The sentence seems too articulate and not effortless. The authors could think to substitute it with a more fluent version, such as “It describes the main in vitro and in vivo activities of SDs having different compositions and various preparation methods, intending to innovate and enhance the release of poorly water-soluble drugs, as well as improving the biological activities of the loaded bioactive compounds (Figure 1).”
Line 55: Figure 1 is just about the applications in vitro and in vivo of the solid dispersion, while for the position chosen along the text it seems that also preparation methods and compositions should me be included in the imagine. I would suggest moving it at the end of the paragraph or editing it.
Line 83-84: About the results, authors just say that the outcomes demonstrate that SD (solid dispersion) was cytotoxic to some cells, mainly PC-3-line ones, while OHPP had no effect on cell viability. But this statement is not complete and clear respect to the information previously reported. Correctly, the cited study says that Niclo-OHPP SD showed higher cytotoxicity than DMSO-assisted niclosamide solutions, mostly on PC-3 cell line, although the OHPP, as the solubilizer, did not contribute to cytotoxicity. In addition, the analysis on OHPP alone is missed.
I would suggest to correct the sentence adding the omitted information.
Line 93-95: The authors report IC50 values of 0.02, 0.08 and 0.03 µg/mL for PTX-OHPP system and 0.008, 0.017 and 0.002 µg/mL for SD system (against HeLa, PC-3, and A549). While the cited study report IC50 values of were 0.02, 0.08, and 0.03 mg/mL of DMSO-assisted PTX solutions and 0.008, 0.017, and 0.002 mg/mL of PTX-OHPP SD.
First, the unit of measurements are different. Authors report a concentration of µg/mL but in the articles the dame data have a unit of mg/mL.
Second, what author express as PTX-OHPP system, seems to be the compound alone (DMSO-assisted solution). In addition the acronym PTX-OHPP should be clarified
All these should be clarified.
Line 104: I would add that the increased cytotoxicity showed by the SD4 and SD9 is respect PTX alone at the same concentration of 20µg/mL.
Line 122-124: “SD of zinc (II)-curcumin complex was developed using polyvinylpyrrolidone (PVP K30) [17], a compound with good proven anticancer activity, but low bioavailability and poor solubility in water, which hinder oral formulation.” Written in this way it seems that the anticancer compound is the PVP K30. I would suggest rephrasing in a clearer way.
Line 303: There is an error in punctuation.
Lines 401-402. To confirm the claims, the author could report the how much more effective the SDs are respect the control and the drug alone. On the other hand, reporting the value of P is not significant.
Lines 419-422. The author should report the respective concentration values
Lines 466. The acronym “OC” does not coincide with any other previously reported
Lines 513. The acronym “BA” does not coincide with any other previously reported
Some acronyms (e.g. line 466 “OC”) are not included in the text or in the abbreviation section below, some others are redundant and often specified long the paper (e.g. line 553, “CM”).
Lines 715. It could be interesting to report in percentage how much the Amalyl1 was able to reduce the glycemia after 3hrs.
Line 879: the doi is not correct.
Line 913: the doi is missing.
Author Response
Line 46: I would move the reference at the end of the sentences, because it is referred both to sucrose, dextrose, and galactose neither to the high thermodynamic stability.
Answer: As suggested, the reference has been moved to the end of these sentences (Line 47).
Line 54-57: The sentence seems too articulate and not effortless. The authors could think to substitute it with a more fluent version, such as “It describes the main in vitro and in vivo activities of SDs having different compositions and various preparation methods, intending to innovate and enhance the release of poorly water-soluble drugs, as well as improving the biological activities of the loaded bioactive compounds (Figure 1).”
Answer: The authors thank the reviewer for this suggestion. This sentence has been rewritten exactly as suggested by him (Lines 54-57), with the only exception of the placement of reference to Figure 1 (see the answer to the subsequent question).
Line 55: Figure 1 is just about the applications in vitro and in vivo of the solid dispersion, while for the position chosen along the text it seems that also preparation methods and compositions should me be included in the imagine. I would suggest moving it at the end of the paragraph or editing it.
Answer: As suggested, Figure 1, which only refers to in vitro and in vivo applications, and not to methods and compositions as well, is now placed just at the end of the paragraph. However, reference to it, “(Figure 1)”, has been moved just after the words “in vitro and in vivo activities” (Line 54) to avoid the reader expecting to see methods and compositions in the figure as well.
Line 83-84: About the results, authors just say that the outcomes demonstrate that SD (solid dispersion) was cytotoxic to some cells, mainly PC-3-line ones, while OHPP had no effect on cell viability. But this statement is not complete and clear respect to the information previously reported. Correctly, the cited study says that Niclo-OHPP SD showed higher cytotoxicity than DMSO-assisted niclosamide solutions, mostly on PC-3 cell line, although the OHPP, as the solubilizer, did not contribute to cytotoxicity. In addition, the analysis on OHPP alone is missed. I would suggest to correct the sentence adding the omitted information.
Answer: This sentence has been corrected as suggested by the reviewer adding the omitted information (Lines 83-85).
Line 93-95: The authors report IC50 values of 0.02, 0.08 and 0.03 µg/mL for PTX-OHPP system and 0.008, 0.017 and 0.002 µg/mL for SD system (against HeLa, PC-3, and A549). While the cited study report IC50 values of were 0.02, 0.08, and 0.03 mg/mL of DMSO-assisted PTX solutions and 0.008, 0.017, and 0.002 mg/mL of PTX-OHPP SD.
First, the unit of measurements are different. Authors report a concentration of µg/mL but in the articles the dame data have a unit of mg/mL.
Second, what author express as PTX-OHPP system, seems to be the compound alone (DMSO-assisted solution). In addition the acronym PTX-OHPP should be clarified
All these should be clarified.
Answer: Regarding the first question, the concentration is expressed in the article in “µg/mL” (Line 95), as shown in excerpt of the cited article (see below):
Regarding the second question, the PTX compound was evaluated either dissolved in the solid dispersion (PTX-OHPP SD) or dissolved in DMSO-assisted solution. It is now written in the text that “The IC50 values of DMSO-assisted PTX solutions against the above tumor cells were 0.02, 0.08 and 0.03 μg/mL, respectively, while PTX-OHPP SD demonstrated the greatest efficacy against cancer cells, with values of 0.008, 0.017 and 0.002 μg/mL, respectively (Lines 94-97).
As for the acronym PTX-OHPP SD, its meaning is now detailed in the text (Lines 90-91).
Line 104: I would add that the increased cytotoxicity showed by the SD4 and SD9 is respect PTX alone at the same concentration of 20µg/mL.
Answer: As suggested by the reviewer, this sentence has been changed to “These results show greater cytotoxicity of SD4 and SD9 compared to PTX alone at the same concentration of 20 µg/mL” (Lines 107-108).
Line 122-124: “SD of zinc (II)-curcumin complex was developed using polyvinylpyrrolidone (PVP K30) [17], a compound with good proven anticancer activity, but low bioavailability and poor solubility in water, which hinder oral formulation.” Written in this way it seems that the anticancer compound is the PVP K30. I would suggest rephrasing in a clearer way.
Answer: The reviewer is right. To avoid any reader’s misunderstanding, this sentence has been divided in two sentences: “In another study on CM, SD of zinc(II)-curcumin complex was developed using polyvinylpyrrolidone (PVP K30) [17]. Metal-curcumin complexes are known to exert anticancer activity, but they have low bioavailability and poor solubility in water, which hinder oral formulations.” (Lines 126-128).
Line 303: There is an error in punctuation.
Answer: The punctuation error has been corrected (Line 307).
Lines 401-402. To confirm the claims, the author could report the how much more effective the SDs are respect the control and the drug alone. On the other hand, reporting the value of P is not significant.
Answer: As suggested, this sentence has been changed to “Specifically, the ABAM-PVP SD prepared by the kneading method had the greatest effect with 62.49% inhibition, while ABAM showed a value of 44.55%.” (Lines 406-408).
Lines 419-422. The author should report the respective concentration values
Answer: As suggested, concentrations of both CM-SD and CM alone are now reported in the text (Lines 423-425).
Lines 466. The acronym “OC” does not coincide with any other previously reported.
Answer: The acronym OC has been replaced by the entire name of the related compound (oleocanthal) (Line 472).
Lines 513. The acronym “BA” does not coincide with any other previously reported.
Answer: The acronym BA has been replaced by the entire name of the related compound (boric acid) (Lines 519 and 521).
Some acronyms (e.g. line 466 “OC”) are not included in the text or in the abbreviation section below, some others are redundant and often specified long the paper (e.g. line 553, “CM”).
Answer: The acronyms have been checked in the list and throughout the text (Lines 472, 519, 521 and 564). However, at the start of any new section the entire word “curcumin” is preferred in the text to make reader’s understanding easier.
Lines 715. It could be interesting to report in percentage how much the Amalyl1 was able to reduce the glycemia after 3hrs.
Answer: As requested, the percentage reduction of glycemia induced by both pure drug and Amaryl1 have been reported (Line 721).
Line 879: the doi is not correct.
Answer: Many thanks. The doi was corrected (Line 884).
Line 913: the doi is missing.
Answer: The doi has been added (Line 919).

Reviewer 2 Report
The most important merit of this manuscript is that it sheds light on the clinical and pre-clinical implications of an important and rather conventional formulation approach. This issue is not much appreciated in pharmaceutical academic circles, and it will help raise awareness. Particularly in the field of Pharmacy education, where curricula are undergoing shifts and revisions around the world toward a more clinically oriented education, such reviews showcase the importance of pharmaceutical technology and formulation science as major constituents of the curriculum even if it gets more clinically oriented.
It is in this regard that the unusual division of subchapters in sections 3 and 4 based on pharmacological activity of the formulated compounds rather than formulation characteristics (as would be conventional in a formulation science article) would come helpful. I know that some fellow formulation scientists might find that bizarre, but being involved in pharmacy education and curriculum discussions, I find this unconventional way of presenting the reviewed info helpful in showcasing the significance of the topic.
This extends also to the field of pharmaceutical academic research, since the people that usually deal with research grant decisions for academia are not formulation scientists but usually come from a clinical or biological background. And they tend to underestimate the importance of anything for which they don’t see a direct clinical application, which puts formulation scientists in academic circles at a disadvantage and is a major factor suppressing formulation research at universities (at least in Western Europe and North America). Such reviews help in ameliorating this problem. All of this comes in addition to it providing a useful overview for an interested researcher in pharmaceutical industry, since the paper is well written and covers many relevant works published in literature.
Other than correcting the typo in the section 4 title "line 433" I don't see any reason not to publish this work.
Author Response
Other than correcting the typo in the section 4 title "line 433" I don't see any reason not to publish this work.
Anwer: Correction performed: In vivo studies on solid dispersions in polymeric matrices.
